# SC3D: Self-conditioned Generative Gaussian Model with 3D-aware Feedback

## Abstract

Existing single image-to-3D creation methods typically involve a two-stage process, first generating multi-view images, and then using these images for 3D reconstruction. However, training these two stages separately leads to significant data bias in the inference phase, thus affecting the quality of reconstructed results. We introduce a unified 3D generation framework, named SC3D, which integrates diffusion-based multi-view image generation and 3D reconstruction through a self-conditioning mechanism. In our framework, these two modules are established as a cyclic relationship so that they adapt to the distribution of each other. During the denoising process of multi-view generation, we feed rendered color images and maps by SC3D itself to the multi-view generation module. This self-conditioned method with 3D aware feedback unites the entire process and improves geometric consistency. Experiments show that our approach enhances sampling quality, and improves the efficiency and output quality of the generation process.

## 1 Introduction

3D content creation from a single image have improved rapidly in recent years with the adoption of large 3D datasets [1, 2, 3] and diffusion models [4, 5, 6]. A body of research [7, 8, 9, 10, 11, 12, 13, 14] has focused on multi-view diffusion models, fine-tuning pretrained image or video diffusion models on 3D datasets to enable consistent multi-view synthesis. These methods demonstrate generalizability and produce promising results. Another group of works [15, 16, 17, 18, 19] propose generalizable reconstruction models, generating 3D representation from one or few views in a feed-forward process. Theses reconstruction models built upon convolutional network or transformer backbone, have led to efficient image-to-3D creation.

Since single-view reconstruction models [15] trained on 3D datasets [1, 20] lack generalizability and often produce blurring at unseen viewpoints, several works [21, 16, 18, 19] extend models to sparse-view input, boosting the reconstruction quality. As shown in Fig. 1, these methods split 3D generation into two stages: multi-view synthesis and 3D reconstruction. By combining generalizable multi-view diffusion models and robust sparse-view reconstruction models, such pipelines achieve high-quality image to 3D generation. However, combining the two independently designed models introduces a significant "data bias" to the reconstruction model. The data bias is mainly reflected in two aspects: **(1) Multi-view bias.** Multi-view diffusion models learn consistency at the image level, struggle to ensure geometric consistency. When it comes to reconstruction, multi-view images that lack geometric consistency affect the subsequent stage. **(2) Limited data for reconstruction model.** Unlike multi-view diffusion models, reconstruction models which are trained from scratch on limited 3D dataset, lacks the generalization ability.

Recent works like IM-3D [22] and VideoMV [23] have attempted to aggregate the rendered views of the reconstructed 3D model into previous-step multi-view synthesis, thus improving the capability

Submitted to 38th Conference on Neural Information Processing Systems (NeurIPS 2024). Do not distribute.

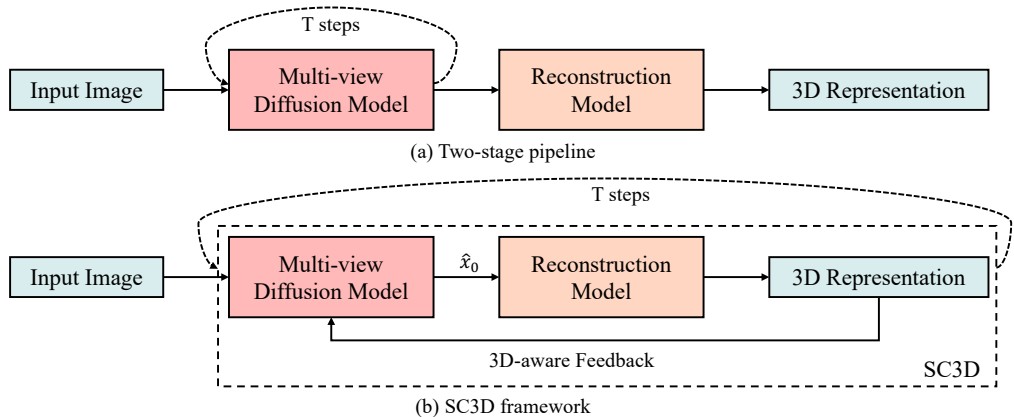

Figure 1: **Concept comparison** between SC3D and previous two-stage methods. Instead of directly combining multi-view diffusion model and reconstruction model, our self-conditioned framework involves joint training of these two models and establish them as a cyclic association. During the denoising process, rendered 3D-aware maps are fed to the multi-view generation module.

and consistency of the generated multi-view images. These methods integrate the aforementioned two stages at the inference phase. But the models at both stages still lack joint training, which prevents the reconstruction model from enhancing its robustness to the generated poor multiviews. Moreover, these test-time aggregating methods cannot directly utilize geometric information such as depth maps, normal maps, or position maps that can also be obtained from the reconstructed 3D. Notably, these explicit 3D aware maps can better guide the multi-view generation.

To address these challenges, we propose a unified single image-to-3D creation framework, named SC3D, which integrates multi-view generation and 3D reconstruction through a self-conditioning mechanism. Our framework involves jointly training the multi-view diffusion model and the reconstruction model. In SC3D, these two modules are established as a cyclic relationship so that they adapt to the characteristics of each other, enabling robust generation at inference. Specifically, during the denoising process, we feed rendered 3D-aware maps from the reconstructed 3D to the multi-view generation module. By leveraging the color maps and spatial canonical coordinates maps from the reconstruction 3D representation as condition, our multi-view diffusion model synthesizes multi-view images that better conform to the actual 3D structure. This self-conditioned framework with 3D aware feedback unites the 3D generation process and enhances the robustness for unseen complex scenes. Experiments on the GSO dataset [24] validate that our SC3D reduces data bias between training and inference, and enhances the overall efficiency and output quality.

Our key contributions are as follows:

- We introduce SC3D, which unifies multi-view generation and 3D reconstruction in a single framework and involves jointly training these two modules, enabling adaption to each other.
- SC3D employs a self-conditioning mechanism with 3D-aware feedback, using rendered 3D-aware maps to guide the multi-view generation, ensuring better geometric consistency and robustness.
- Experiments show that SC3D significantly reduces data bias, improves the quality of 3D reconstruction, and enhances overall efficiency in creating 3D content from a single image.

## 2 Related Work

**Image/Video Diffusion for Multi-view Generation**    Diffusion models [25, 26, 27, 28, 29, 30, 31, 32, 33, 34] have demonstrated their powerful generative capabilities in image and video generation fields. Current research [7, 8, 9, 10, 11, 12, 13, 14, 35] fine-tunes pretrained image/video diffusion models on 3D datasets like Objaverse [1] and MVImageNet [20]. Zero123 [7] introduces relative view condition to image diffusion models, enabling novel view synthesis from a single image and preserving generalizability. Based on it, methods like SyncDreamer [9], ConsistNet [36] and EpiDiff [11] design attention modules to generate consistent multi-view images. These methods fine-

tuned from image diffusion models produce generally promising results. By considering multi-view images as consecutive frames of a video (e.g., orbiting camera views), it naturally leads to the idea of applying video generation models to 3D generation [13]. However, since the diffusion model is not explicitly modeled in 3D space, the generated multi-view images often struggle to achieve consistent and robust details.

**Image to 3D Reconstruction**   Recently, the task of reconstructing 3D objects has evolved from traditional multi-view reconstruction methods [37, 38, 39, 40] to feed-forward reconstruction models [15, 41, 42, 16, 17, 18, 19]. Ultilizing one or few shot as input, these highly generalizable reconstruction models synthesize 3D representation, enabling the rapid generation of 3D objects. LRM [15] proposes a transformer-based model to effectively map image tokens to 3D triplanes. Instant3D [21] further extends LRM to sparse-view input, significantly boosting the reconstruction quality. LGM [16] and GRM [17] replace the triplane representation with 3D Gaussians [40] to enjoy its superior rendering efficiency. CRM [18] and InstantMesh [19] optimize on the mesh representation for high-quality geometry and texture modeling. These reconstrucion models built upon convolutional network architecture or transformer backbone, have led to efficient image-to-3D creation.

**Pipelines of 3D Generation**   Early works propose to distill knowledge of image prior to create 3D models via Score Distillation Sampling (SDS) [43, 44, 45], limited by the low speed of per-scene optimization. Several works [9, 11, 14, 22] fine-tune image diffusion models to generate multi-view images, which are then utilized for 3D shape and appearance recovery with traditional reconstruction methods [46, 40]. More recently, several works [21, 16, 18, 19, 23] involve both multi-view diffusion models and feed-forward reconstruction models in the generation process. Such pipelines attempt to combine the processes into a cohesive two-stage approach, thus achieving highly generalizable and high-quality single-image to 3D generation. However, due to the lack of explicit 3D modeling, the results generated by the multi-view diffusion model cannot guarantee strong consistency, which will lead to data deviation for the reconstructed model between the testing phase and the training phase. Compared to them, we propose a unified pipeline, integrating the two stages through a self-conditioning mechanism at the training stage, with 3D aware feedback for high consistency.

# 3   Method

Given a single image, SC3D aims to generate multiview-consistent images with a reconstructed 3D Gaussion model. To reduce the data bias and improve robustness of the generation, we propose SC3D, a unified 3D generation framework which integrates multi-view synthesis and 3D reconstruction through a self-conditioning mechanism. As illustrated in Fig. 2, the proposed framework involves a video diffusion model (SVD [32]) as multi-view generator (refer to Section 3.1) and a feed-forward reconstruction model to recover a 3D Gaussian Splatting (refer to Section 3.2. Moreover, we introduce a self-conditioning mechanism, feeding the 3D-aware information obtained from the reconstruction module back to the multi-view generation process (refer to Section 3.3). The 3D-aware denoising sampling strategy iteratively refines the multi-view images and the 3d model, thus enhancing the final production.

## 3.1   Video Diffusion Model as Multiview Generator

Recent video diffusion models such as those in [13, 34] have demonstrated a remarkable capability to generate 3D-aware videos by scaling up both the model and dataset. Our research employs the well-known Stable Video Diffusion (SVD) Model, which generates videos from image input. Formally, given an image $I \in \mathbb{R}^{3 \times h \times w}$, the model is designed to generate a video $V \in \mathbb{R}^{f \times 3 \times h \times w}$. Further details about SVD can be found in Appendix A.1.

We enhance the video diffusion model with camera control $c$ to generate images from different viewpoints. Traditional methods encode camera positions at the frame level, which results in all pixels within one view sharing the same positional encoding [47, 13]. Building on the innovations of previous work [11, 35], we integrate the camera condition $c$ into the denoising network by parameterizing the rays $\mathbf{r} = (o, o \times d)$. Specifically, we use two-layered MLP to inject Plücker ray embeddings for each latent pixel, enabling precise positional encoding at the pixel level. This approach allows for more detailed and accurate 3D rendering, as pixel-specific embedding enhances the model's ability to handle complex variations in depth and perspective across the video frames.

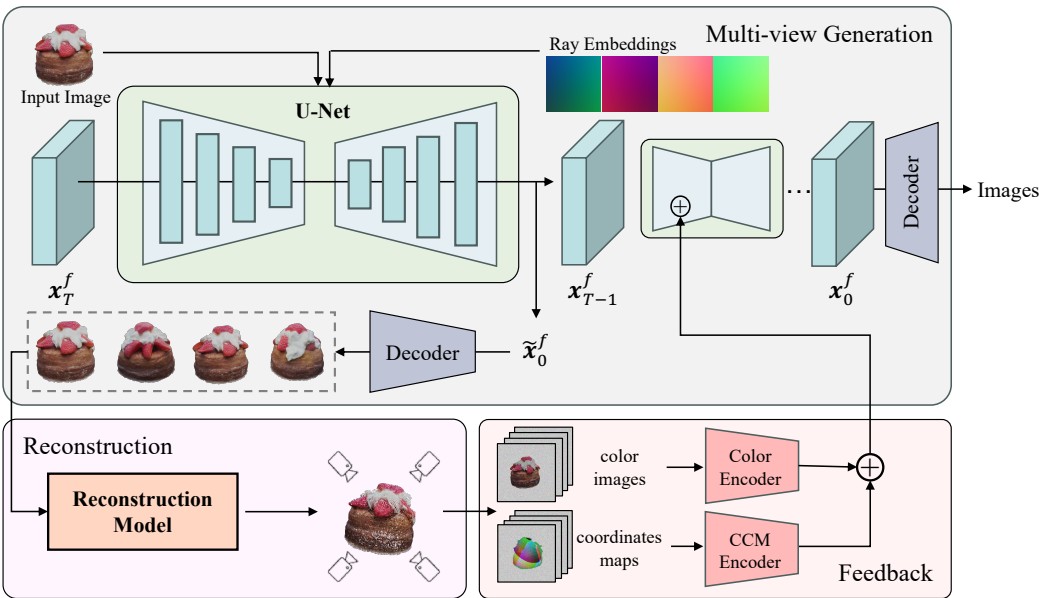

Figure 2: **Overview of SC3D.** We adopt a video diffusion model as the multi-view generator by incorporating the input image and relative camera poses. In the denoising sampling loop, we decode the predicted $\widetilde{\mathbf{x}}_0^f$ to noise-corrupted images, which are then used to recover 3D representation by a feed-forward reconstruction model. Then the rendered color images and coordinates maps are encoded and fed into the next denoising step. At inference, the 3D-aware denoising sampling strategy iteratively refines the images by incorporating feedback from the reconstructed 3D into the denoising loop, enhancing multi-view consistency and image quality.

In our framework, unlike existing two-stage methods, our multi-view diffusion model does not complete multiple denoising steps independently. In contrast, in the denoising sampling loop, we obtain the straightly predicted $\widetilde{\mathbf{x}}_0^f$ at the current timestep, which will be used for subsequent 3D reconstruction. Then we use rendered 3d-aware view maps as conditions to guide the next denoising step. Therefore, at each sampling step, we do the reparameterization of the output from the denoising network $F_\theta$ to convert it into $\widetilde{\mathbf{x}}_0^f$. Taking a single view as an example, we processes the denoised image $c_{\text{in}}(\sigma)\mathbf{x}$ and the associated noise level $c_{\text{noise}}(\sigma)$, which $\sigma$ indicates the standard deviation of the noise. The reparameterization is formulated as follows:

$$\tilde{\mathbf{x}}_0 = c_{\text{skip}}(\sigma)\mathbf{x} + c_{\text{out}}(\sigma)F_\theta(c_{\text{in}}(\sigma)\mathbf{x}; c_{\text{noise}}(\sigma)). \tag{1}$$

The above operation process adjusts the output of $F_\theta$ to $\widetilde{\mathbf{x}}_0^f$, which will be decoded into images and passed to the subsequent 3D reconstruction module.

## 3.2 Feed-Forward Reconstruction Model

In the SC3D framework, the feed-forward reconstruction model is designed to recover 3D models from pre-generated multi-view images, which can be images decoded from straightly predicted $\widetilde{\mathbf{x}}_0^f$, or completely denoised images. We utilize Large Multi-View Gaussian Model (LGM) [16] $\mathcal{G}$ as our reconstruction module due to its real-time rendering capabilities that benefit from 3D representation of Gaussian Splatting. This method integrates seamlessly with our jointly training framework, allowing for quick adaptation and efficient processing.

We pass four specific views from the reparameterized output $\widetilde{\mathbf{x}}_0^f$ to the Large Gaussian Model (LGM) for 3D Gaussian Splatting reconstruction. To enhance the performance of LGM, particularly its sensitivity to different noise levels $c_{\text{noise}}(\sigma)$ and image details, we introduce a zero-initialized time embedding layer within the original U-Net structure of the LGM. This innovative modification enables the LGM to dynamically adapt to the diverse outputs that arise at different stages of the

144 denoising process, thereby substantially improving its capacity to accurately reconstruct 3D content
145 from images that have undergone partial denoising.

146 The loss function employed for the fine-tuning of the LGM is articulated as follows:

$$\mathcal{L}_{\mathcal{G}} = \mathcal{L}_{\text{rgb}}(\mathbf{x}_0, \mathcal{G}(\tilde{\mathbf{x}}_0, c_{\text{noise}}(\sigma))) + \lambda \mathcal{L}_{\text{LPIPS}}(\mathbf{x}_0, \mathcal{G}(\tilde{\mathbf{x}}_0, c_{\text{noise}}(\sigma))). \qquad (2)$$

147 where we have utilized the mean square error loss $\mathcal{L}_{\text{rgb}}$ for the color channel and a VGG-based
148 perceptual loss $\mathcal{L}_{\text{LPIPS}}$[43] for the LPIPS term. In practical applications, the weighting factor $\lambda$ is
149 conventionally set to 1.

150 Additionally, to maintain the model's reconstruction capability for normal images, we also input the
151 model without adding noise and calculate the corresponding loss. In this case, we set $c_{\text{noise}}(\sigma)$ to 0.

## 3.3 3D-Aware Feedback Mechanism

153 As shown in Fig. 2, we adopt a 3D-aware feedback mechanism that involves the rendered color
154 images and geometric maps produced by our reconstruction module in a denoising loop to further
155 improve the multi-view consistency of the resulting images and facilitate cyclic adaptation of the
156 two stages. Instead of integrating multi-view generation and 3D reconstruction at the inference stage
157 using re-sampling strategy [22, 23], we propose to train these two modules jointly to support more
158 informative feedback. Specifically, in addition to the rendered color images, our flexible framework
159 is able to derive additional geometric features to guide the generation process, which brings guidance
160 of more explicit 3D information to multi-view generation.

161 In practice, we obtain color images and canonical coordinates maps [48] from the reconstructed 3D
162 model, and utilize them as condition to guide the next denoising step of multi-view generation. We
163 use position maps instead of depth maps or normal maps as the representative of geometric maps
164 because canonical coordinate maps record the vertex coordinate values after normalization of the
165 overall 3D model, rather than the normalization of the relative self-view (such as depth maps). This
166 operation enables the rendered maps to be characterized as cross-view alignment, providing the strong
167 guidance of more explicit cross-view geometry relationship. The details of canonical coordinates
168 map can be found in Appendix A.2.

169 We adopt a 3D-aware self-conditioning [49] training and inference strategy that leverages reconstruc-
170 tion stage results to enhance multi-view consistency and the quality of generated images. During
171 training, the original denoising network $F_\theta(\mathbf{x}; \sigma)$ is augmented with a 3D-aware feedback denoising
172 network $F_\theta(\mathcal{G}(\tilde{\mathbf{x}}_0); \sigma)$, where $\mathcal{G}(\tilde{\mathbf{x}}_0)$ is the output of the LGM reconstruction.

173 To encode color images and coordinates maps into the denoising network of multi-view generation
174 module, we design two simple and lightweight encoders for color images and coordinates maps using
175 a series of convolutional neural networks, like T2I-Adapter [50]. The encoders are composed of four
176 feature extraction blocks and three downsample blocks to change the feature resolution, so that the
177 dimension of the encoded features is the same as the intermediate feature in the encoder of U-Net
178 denoiser. The extracted features from the two conditional modalities are then added to the U-Net
179 encoder at each scale.

180 **Training Strategy**    As illustrated in Algorithm 1, to train a 3D-aware multi-view generation network,
181 we use the rendered maps by the 3D reconstruction module as the self-conditioning input. In practice,
182 we randomly use this self-conditioning mechanism with a probability of 0.5. When not using the 3D
183 reconstruction result, we set $\mathcal{G}(\tilde{x}_0) = 0$ as the input. This probabilistic approach ensures balanced
184 learning, allowing the model to effectively incorporate 3D information without over-reliance on it.

**Algorithm 1** Training SC3D with the self-conditioned strategy.

```python
def train_loss(x, cond_image):
    """Returns the loss on a training example x."""
    # Sample sigma from a log-normal distribution
    sigma = log_normal(P_mean, P_std)

    # Reparameterize sigma to obtain conditioning parameters
    c_in, c_out, c_skip, c_noise, lambda_param = reparameterizing(sigma)

    # Add noise to input data
    noise_x = x + sigma * normal(mean=0, std=1)
    input_x = c_in * noise_x

    # Initial prediction without self-conditioning
    self_cond = None
    F_pred = net(input_x, c_noise, cond_image, self_cond)
    pred_x = c_out * F_pred + c_skip * noise_x

    # Update self_cond using the reconstruction model
    self_cond = recon_model(pred_x, c_noise)

    # Use rendered maps as condition and denoise
    if self_cond and np.random.uniform(0, 1) > 0.5:
        F_pred = net(input_x, t, cond_image, self_cond.detach())
        pred_x = c_out * F_pred + c_skip * noise_x

    # Compute loss
    loss = lambda_param * (pred_x - target) ** 2
    recon_loss = recon_loss_fn(self_cond, x)

    return loss.mean() + recon_loss
```

**Inference/sampling strategy**   At the inference stage, as shown in Algorithm 2, the 3D feedback $\mathcal{G}(\tilde{\mathbf{x}}_0)$ is initially set to 0. At each timestep, this feedback is updated with the previous reconstruction result $\mathcal{G}(\tilde{\mathbf{x}}_0)$. This iterative process refines the 3D representation, ensuring each frame benefits from prior reconstructions, leading to higher quality and more consistent 3D-aware images.

**Algorithm 2** Sampling algorithm of SC3D.

```python
def generate(sigmas, cond_image):
    self_cond = None
    x_T = normal(mean=0, std=1)   # Initialize latent variable with Gaussian noise
    for sigma in sigmas:
        # Reparameterize sigma to obtain conditioning parameters
        c_in, c_out, c_skip, c_noise, lambda_param = reparameterizing(sigma)

        # Add noise to the latent variable
        noise_x = x_T + sigma * normal(mean=0, std=1)
        input_x = c_in * noise_x

        # Generate prediction
        F_pred = net(input_x, t, cond_image, self_cond)
        pred_x = c_out * F_pred + c_skip * noise_x

        # Update self_cond using the reconstruction model
        self_cond = recon_model(pred_x, c_noise)

    return pred_x
```

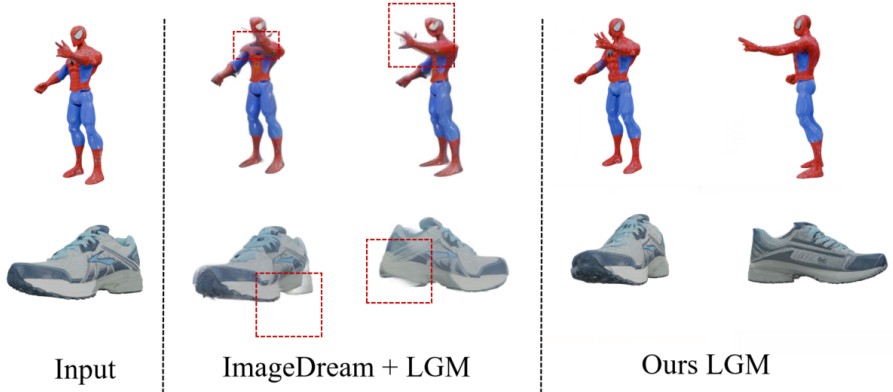

Input        ImageDream + LGM        Ours LGM

Figure 3: Qualitative comparison with ImageDream-LGM and Our LGM.

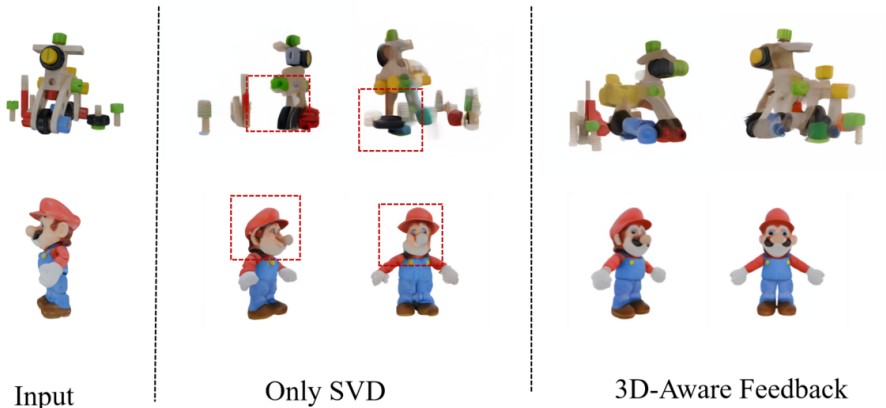

Input        Only SVD        3D-Aware Feedback

Figure 4: Qualitative comparison with no-feedback and 3d-aware feedback.

## 4 Experiments

We focus on 3D asset content synthesis, training our model on the G-Objaverse [1, 51] dataset and the LVIS subset of Objaverse, which consists of 300K high-quality 3D objects and is widely used in 3D generation. We evaluate SC3D on the Google Scanned Object (GSO) dataset [24], which consists of approximately 1,000 scanned models, and we randomly select 100 samples for comparison. We adopt TripoSR[42], SyncDreamer[9], SV3D[13], ImageDream [8] combined with LGM [16] as the baseline approach [16] and VideoMV[23] as baseline methods. For each baseline, we report PSNR, SSIM, and LPIPS metrics.

### 4.1 Comparison results

For LGM, we utilize the official LGM single-image generation pipeline, which employs ImageDream [52] to transition from a single image input to multiple images. However, the conical coordinate system employed by ImageDream complicates the direct evaluation of the output. To address this, we use the official code to test on the GSO dataset, followed by manual calibration to assess the generated quality, as illustrated in Fig. 3. The misalignment between the two stages of ImageDream and LGM often results in generated models with blurred linear edges and geometric ambiguities. Nonetheless, our LGM, enhanced by a feedback mechanism, demonstrates significantly improved geometric and texture quality, producing results that closely approximate reality.

As illustrate in 6, We find that although it can generate very continuous frames, the generated content tends to deviate from the given input image. This results in sub-optimal performance in

| Method | Resolution | PSNR↑ | SSIM↑ | LPIPS↓ |
|---|---|---|---|---|
| TripoSR | $256 \times 256$ | 18.481 | 0.8506 | 0.1357 |
| SyncDreamer | $256 \times 256$ | 20.056 | 0.8163 | 0.1596 |
| SV3D | $576 \times 576$ | 21.042 | 0.8497 | 0.1296 |
| VideoMV(SD) | $256 \times 256$ | 17.459 | 0.806 | 0.1446 |
| VideoMV(GS) | $256 \times 256$ | 17.577 | 0.807 | 0.1454 |
| SC3D (SVD) | $512 \times 512$ | 21.625 | 0.9045 | 0.1011 |
| SC3D (GS) | $512 \times 512$ | **21.761** | **0.9094** | **0.0991** |

Table 1: Comparison of performance metrics across different models and configurations.

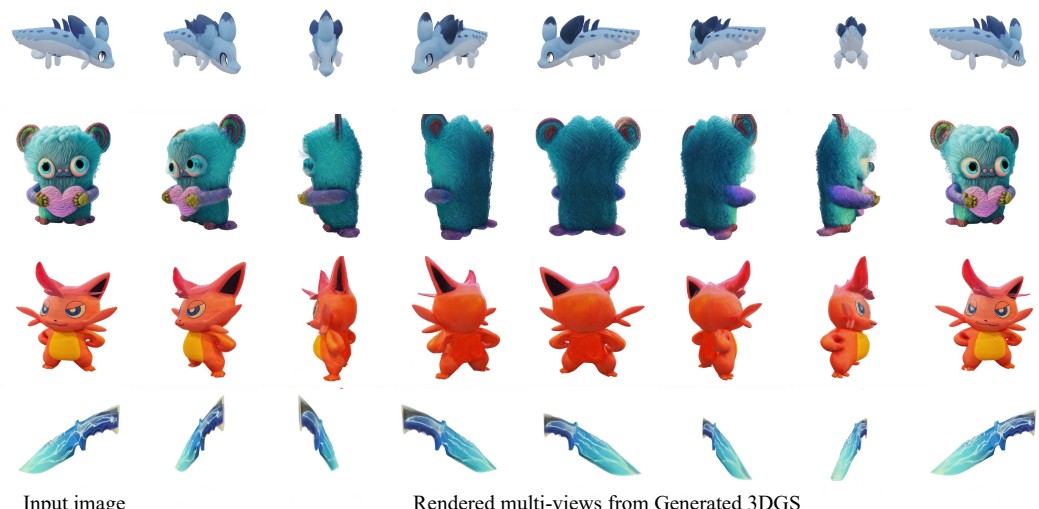

Input image      Rendered multi-views from Generated 3DGS

Figure 5: Out of distribution testing results.

the reconstruction metric. Additionally, VideoMV training the LGM separately with noisy images deteriorates, resulting in a visually noticeable reduction in its ability to generate texture details.

## 4.2 Ablation study

To validate the effectiveness of the proposed SC-3D framework, we conducted a series of ablation studies comparing PSNR, SSIM, and LPIPS metrics for different configurations (Table 2). We start with the base video diffusion model we trained, We then introduced 3D coordinates map feedback and RGB texture feedback from the reconstruction model to the diffusion model, which improved geometric consistency and texture detail across views. Combining both feedback mechanisms in the SVD + 3D-aware Feedback configuration resulted in the best performance, demonstrating significant improvements in the final 3D reconstruction quality by enhancing both geometric consistency and texture detail preservation.

| Method | Variant | PSNR ↑ | SSIM ↑ | LPIPS ↓ |
|---|---|---|---|---|
| SVD | SVD | 20.038 | 0.8745 | 0.1253 |
| | GS | 20.549 | 0.8651 | 0.1183 |
| SVD + Coordinates Map Feedback | SVD | 21.021 | 0.8973 | 0.1110 |
| | GS | 21.325 | 0.8937 | 0.1092 |
| SVD + 3D-aware Feedback | SVD | 21.752 | **0.9122** | 0.0993 |
| | GS | **21.761** | 0.9094 | **0.0991** |

Table 2: Performance metrics of different feedback mechanisms.

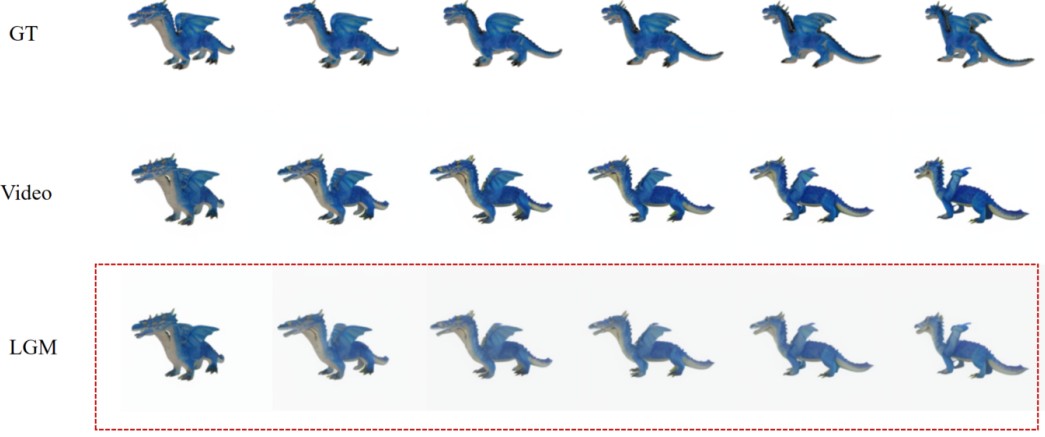

Figure 6: The Generation Example of VideoMV

We also demonstrate the impact of incorporating feedback mechanisms on the two models, as shown in Table 3. It can be observed that when no feedback mechanism is used, there is a significant discrepancy between the two models' modalities, which leads to a degradation in their combined performance.

| Method | Delta PSNR | Delta SSIM | Delta LPIPS |
|---|---|---|---|
| SVD | 0.511 | 0.0094 | 0.0070 |
| SVD + Coordinates Map Feedback | 0.304 | 0.0036 | 0.0018 |
| SVD + 3D-aware Feedback | 0.009 | 0.0028 | 0.0002 |

Table 3: The absolute differences in performance metrics between GS and SVD generation results..

### 4.3 Limitations

Current models utilize Gaussian splatting as a 3D representation, mapping and rendering coordinates to textures for feedback. Although algorithms for converting Gaussian Splatting to meshe are under development, achieving high quality in converting Gaussian models to general meshes remains challenging. Directly employing a NeRF-based feed-forward model during the training process significantly reduces training speed due to the computational demands of volumetric rendering. Our model currently lacks the ability to generalize to the scene level, a limitation we intend to address in future research.

## 5 Conclusion

In this paper, we introduce SC3D, a unified framework for 3D generation from a single image that integrates multi-view image generation and 3D reconstruction through a self-conditioning mechanism. By establishing a cyclic relationship between these two stages, our approach effectively mitigates the data bias encountered in traditional methods. The self-conditioned method with 3D-aware feedback enhances geometric consistency throughout the generation process.

Our experiments demonstrate that SC3D not only improves the quality and efficiency of the generation process but also achieves superior geometric consistency and detail in the reconstructed 3D models. By jointly training the multi-view diffusion model and the reconstruction model, SC3D adapts to the inherent biases of each stage, resulting in more robust and accurate outputs.

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

## A  Technical Details

### A.1  Video model finetuning

Based on the approach outlined in [32], the generation process employs the EDM framework[53]. Let $p_{\text{data}}(\mathbf{x}_0)$ represent the video data distribution, and $p(\mathbf{x}; \sigma)$ be the distribution obtained by adding Gaussian noise with variance $\sigma^2$ to the data. For sufficiently large $\sigma_{\max}$, $p(x; \sigma_{\max}^2)$ approximates a normal distribution $\mathcal{N}(0, \sigma_{\max}^2)$. Diffusion models (DMs) leverage this property and begin with high variance Gaussian noise, $x_M \sim \mathcal{N}(0, \sigma_{\max}^2)$, and then iteratively denoise the data until reaching $\sigma_0 = 0$.

In practice, this iterative refinement process can be implemented through the numerical simulation of the Probability Flow ordinary differential equation (ODE):

$$d\mathbf{x} = -\dot{\sigma}(t)\sigma(t)\nabla_{\mathbf{x}} \log p(\mathbf{x}; \sigma(t))\, dt \tag{3}$$

where $\nabla_{\mathbf{x}} \log p((\mathbf{x}; \sigma)$ is called as score function.

DM training is to learn a model $s_\theta(\mathbf{x}; \sigma)$ to approximate the score function $\nabla_{\mathbf{x}} \log p((\mathbf{x}; \sigma)$. The model can be parameterized as:

$$\nabla_{\mathbf{x}} \log p((\mathbf{x}; \sigma) \approx s_\theta((\mathbf{x}; \sigma) = \frac{D_\theta(\mathbf{x}; \sigma) - \mathbf{x}}{\sigma^2}, \tag{4}$$

where $D_\theta$ is a learnable denoiser that aims to predict ground truth $\mathbf{x}_0$.

The denoiser $D_\theta$ is trained via denoising score matching (DSM):

$$\mathbb{E}_{\mathbf{x}_0 \sim p_{\text{data}}(\mathbf{x}_0),(\sigma,n)\sim p(\sigma,n)} \left[ \lambda_\sigma \| D_\theta(\mathbf{x}_0 + n; \sigma) - \mathbf{x}_0 \|_2^2 \right], \tag{5}$$

where $p(\sigma, n) = p(\sigma)\mathcal{N}(n; 0, \sigma^2)$, $p(\sigma)$ is a distribution over noise levels $\sigma$, $\lambda_\sigma$ is a weighting function. The learnable denoiser $D_\theta$ is parameterized as:

$$D_\theta(\mathbf{x}; \sigma) = c_{\text{skip}}(\sigma)\mathbf{x} + c_{\text{out}}(\sigma)F_\theta(c_{\text{in}}(\sigma)\mathbf{x}; c_{\text{noise}}(\sigma)), \tag{6}$$

where $F_\theta$ is the network to be trained.

We sample $\log \sigma \sim \mathcal{N}(P_{\text{mean}}, P_{\text{std}}^2)$, with $P_{\text{mean}} = 1.0$ and $P_{\text{std}} = 1.6$. Then we obtain all the parameters as follows:

$$c_{\text{in}} = \frac{1}{\sqrt{\sigma^2 + 1}} \tag{7}$$

$$c_{\text{out}} = \frac{-\sigma}{\sqrt{\sigma^2 + 1}} \tag{8}$$

$$c_{\text{skip}}(\sigma) = \frac{1}{\sigma^2 + 1} \tag{9}$$

$$c_{\text{noise}}(\sigma) = 0.25 \log \sigma \tag{10}$$

$$\lambda(\sigma) = \frac{1 + \sigma^2}{\sigma^2} \tag{11}$$

We fine-tune the network backbone $F_\theta$ on multi-view images of size $512 \times 512$. During training, for each instance in the dataset, we uniformly sample 8 views and choose the first view as the input view. view images of size $512 \times 512$.

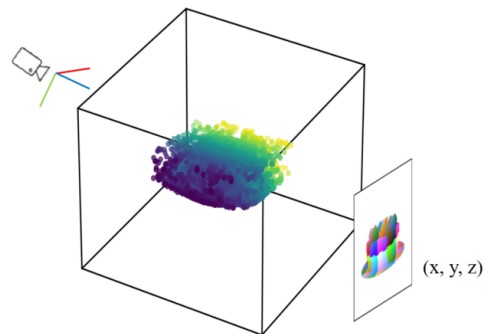

Figure 7: The projection process of coordinates map.

## A.2 Coordinates Map

In conditional control models such as ControlNet[54], T2IAdapter, when depth maps are used as input, their range needs to be normalized to the [0, 1] interval, typically using the formula: $(p - p_{mean})/(p_{max} - p_{min})$. However, this normalization process may introduce scale ambiguity, which can affect the multi-view generation performance. To avoid the issues caused by normalization, we use coordinate maps. Coordinate maps transform the depth value $d$ to a common world coordinate system using the camera's intrinsic and extrinsic parameters, represented as $(X, Y, Z)$. The transformation formula is:

$$\begin{pmatrix} X \\ Y \\ Z \end{pmatrix} = K^{-1} \cdot \begin{pmatrix} u \\ v \\ 1 \end{pmatrix} \cdot d$$

where $(u, v)$ are the pixel coordinates, $d$ is the corresponding depth value, and $K$ is the camera intrinsic matrix.

## A.3 3D Feedback

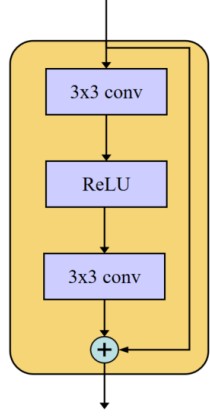

Figure 8: Architecture of the residual block used in feedback stage.

| Input | inp $\in \mathbb{R}^{3 \times 512 \times 512}$ |
|---|---|
| PixelUnshuffle [55] | $192 \times 64 \times 64$ |
| ResBlock $\times 3$ | $320 \times 64 \times 64$ |
| ResBlock $\times 3$ | $640 \times 32 \times 32$ |
| ResBlock $\times 3$ | $1280 \times 16 \times 16$ |
| ResBlock $\times 3$ | $1280 \times 8 \times 8$ |

Table 4: The detailed structure of all layers in the feedback injection network.

With reference to Section 3.3 in the main paper, Fig. 8 and Table 4 provide a detailed illustration of the feedback injection netwrok. We use two networks to inject the coordinates map and RGB texture map feedback into the score function. Each network consists of four feature extraction blocks and three downsample blocks to adjust the feature resolution. The reconstruction coordinates map and

RGB texture map initially have a resolution of $512 \times 512$. We employ the pixel unshuffle operation to downsample these maps to $64 \times 64$.

At each scale, three residual blocks[56] are used to extract the multi-scale feedback features, denoted as $F_P = \{F_p^1, F_p^2, F_p^3, F_p^4\}$ and $F_T = \{F_t^1, F_t^2, F_t^3, F_t^4\}$ for the coordinates map and RGB texture map, respectively. These feedback features match the intermediate features $F_{\text{enc}} = \{F_{\text{enc}}^1, F_{\text{enc}}^2, F_{\text{enc}}^3, F_{\text{enc}}^4\}$ in the encoder of the UNet denoiser. The feedback features $F_P$ and $F_T$ are added to the intermediate features $F_{\text{enc}}$ at each scale as described by the following equations:

$$\mathbf{F}_p = \mathcal{F}^0(P) \tag{12}$$

$$\mathbf{F}_t = \mathcal{F}^1(T) \tag{13}$$

$$\mathbf{F}_{\text{enc}}^i = \mathbf{F}_{\text{enc}}^i + \mathbf{F}_p^i + \mathbf{F}_t^i, \quad i \in \{1, 2, 3, 4\} \tag{14}$$

where $P$ represents the coordinates map feedback input, and $T$ represents the RGB texture feedback input. $\mathcal{F}^0$ and $\mathcal{F}^1$ denote the functions of the feedback inject network applied to the coordinates map and RGB texture map, respectively.

## B  Training Details and Experimental Settings

**Implementation** As illustrate in Table 5, all models are trained for 30,000 iterations using 8 A100 GPUs with a total batch size of 32. We clip the gradient with a maximum norm of 1.0. We use the AdamW optimizer with a learning rate of $1 \times 10^{-5}$ and employ FP16 mixed precision with DeepSeed[57] with Zero-2 for efficient training. We adjust the cameras in each batch so that the initial input view consistently represents the reference frame, using an identity rotation matrix and a fixed translation for alignment.

The inference settings are shown in Table 6.

| Hyperparameter | SVD (1.8 B) | LGM (424M) |
|---|---|---|
| **Training** | | |
| Optimizer | AdamW | AdamW |
| Learning rate | 1e-5 | 1e-5 |
| Batch size per GPU | 4 | 4 |
| # training steps | 40k | 40k |
| # GPUs | 8 | 8 |
| Training time (days) | 4 | 4 |
| Input Resolution | $8 \times 512 \times 512 \times 3$ | $4 \times 256 \times 256 \times 3$ |
| Output Resolution | $8 \times 512 \times 512 \times 3$ | $- \times 512 \times 512 \times 3$ |
| **Diffusion setup** | | |
| $P_{\text{mean}}$ | 1.0 | - |
| $P_{\text{std}}$ | 1.6 | - |

Table 5: Hyperparameters for the training stage.

| Hyperparameter | SC3D | VideoMV | SV3D | SyncDreamer |
|---|---|---|---|---|
| **Sampling parameters** | | | | |
| Sampler | Euler | DDIM | Euler | DDIM |
| steps | 25 | 50 | 50 | 50 |
| cfg gudiance | $1.0 \sim 3.0$ | 6.0 | 6.0 | 2.0 |

Table 6: Hyperparameters for the inference stage.

## C  Additional Visualization Results

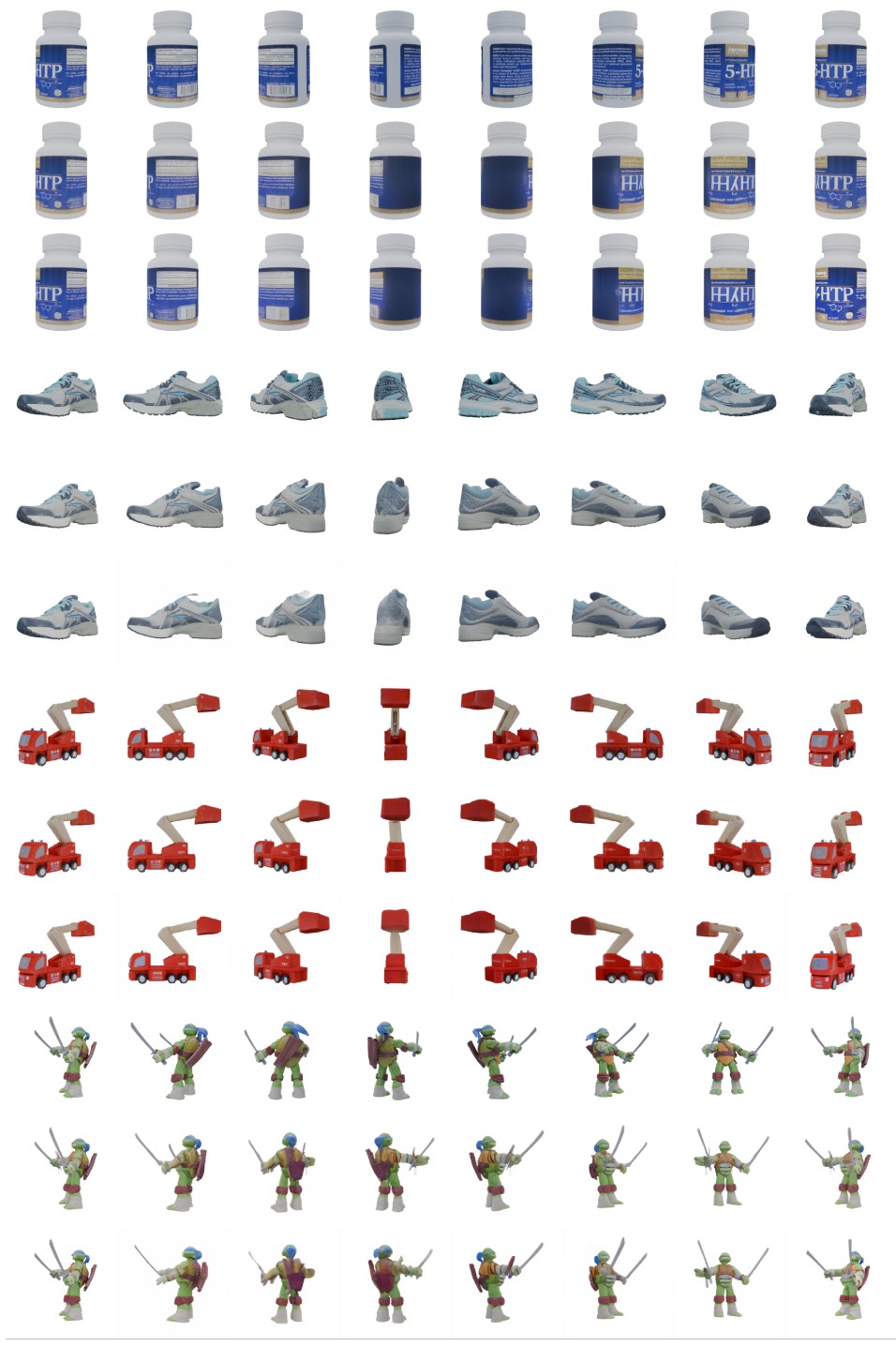

Figure 9: Visualization results generated by our SC3D. For each sample (3 rows), the 1st row is ground truth, 2nd row is the generated multi-view images, while 3rd row is the rendered views from reconstructed 3DGS. For each row, the first image is the input image.

