# OpenReview forum: "SC3D: Self-conditioned Generative Gaussian Model with 3D-aware Feedback"
_NeurIPS.cc/2024/Conference — Submitted to NeurIPS 2024_

### Official Review · Reviewer_DdFX · 2024-07-06

**Soundness:** 3
**Presentation:** 3
**Contribution:** 3
**Rating:** 5
**Confidence:** 4

**Summary:**

This paper (SC3D) proposes a single image-to-3D reconstruction method. It combines multi-view diffusion model and a 3D reconstruction model, and uses the 3D reconstruction results as self condition to improve the multi-view generation process. The motivation of the proposed method is to improve the geometric consistency previous single image reconstruction pipeline, namely first generate multi-view images then perform sparse view reconstruction. The core idea proposed in the paper, 3D-aware feedback, is reasonable and also appear in concurrent works IM-3D and VideoMV. Several ablation studies need to be included to prove that the proposed 3D-feedback (including RGB and coordinate maps) are improving the reconstruction quality. Authors also need to justify more about the contribution w.r.t. related work VideoMV. Furthermore, there is still space to improve the readability in the submission.

**Strengths:**

Major:
- The idea of using 3D reconstruction rendering as condition to improve the geometry consistency of multi-view diffusion models is reasonable.
- The experiments are comprehensive. The results demonstrate that the proposed feedback mechanism is solid in the multi-view reconstruction approach.

**Weaknesses:**

- Claim about key contributions: the 3D-feedback idea appears already in VideoMV. Since the VideoMV is already available on Arxiv in March, authors need to justify more clearly about the difference and contribution w.r.t. VideoMV.
- lacks generalization results: the method is evaluated on google scan objects, which is standard. However, i am curious to see if the approach generalizes to real world images.
- lacks one ablation: SVD+RGBs feedback, which is missing in Tab. 2 and Tab. 3.
- the readability of the Alg.1 and Alg.2 can be improved. Currently it is too specific and looks like python program. A more abstract algorithm is expected in a scientific paper.
- A typo: in line 213, after comma, "we" instead of "We" (wrong capitalized "W")

**Questions:**

- Why there is no SVD+RGB feedback in the ablation study? I think that is important to see how does the coordinates map contribute compared to multi-view RGB images as feedback.

**Limitations:**

- No obvious limitations are found in the proposed method.

---

> ### Author Rebuttal · Authors · 2024-08-07
>
> ***1. Claim about key contributions: the 3D-feedback idea appears already in VideoMV. Since the VideoMV is already available on Arxiv in March, authors need to justify more clearly about the difference and contribution w.r.t. VideoMV.***
>
> We will more clearly justify the difference with VideoMV in our revised paper.
> - VideoMV aims to generate **multi-view consistent** images through 3D-aware denoising sampling. However, on the overall image-to-3D pipeline, its (a) lacking joint training and (b) inability to use geometric information hinder its capacity to fully leverage 3D-aware knowledge and unify the two stages. Moreover, this method fails to address the **"data bias"** between multi-view generation and 3D reconstruction. While its 3D-aware sampling method, employed in the inference phase, can improve multi-view consistency, it also introduces **biased information from reconstructed 3D models**, resulting in outputs that are misaligned with the input image (see **Fig.3 and Fig.4 in our attached PDF**).
> - Our approach begins with a two-stage image-to-3D generation pipeline, aiming to address the **"data bias"** problem in the two stages, and thus propose a unified self-conditioning framework to achieve high-quality 3D generation. We utilize both geometry and appearance information from the reconstructed results, incorporating joint training of the two stages. **The feedback loops in both the training and inference stage** help reduce data bias and generate 3d assets with high-quality geometry and textures adhere to the input image.
>
> ***2. Lacks generalization results: the method is evaluated on google scan objects, which is standard. However, i am curious to see if the approach generalizes to real world images.***
>
> We provide the generalization results in **Fig.2 of our attached PDF**. Our SC3D can also generate high-quality 3D models from out-of-distribution images and real world images.
>
> ***3. lacks one ablation: SVD+RGBs feedback, which is missing in Tab. 2 and Tab. 3.***
>
> We acknowledge the importance of ablating coordinate map feedback. As demonstrated in **Fig. 6 and Tab. 1 in our attached PDF**, the reconstructed results in the 4th column show that relying solely on the color map leads to poor geometric quality in fine details. The combination of RGB and coordinates map feedback  provides the best results by enhancing both geometry and texture quality, showcasing superior performance.
>
> ***4. The readability of the Alg.1 and Alg.2 can be improved. Currently it is too specific and looks like python program. A more abstract algorithm is expected in a scientific paper.***
>
> We revise the two algorithms and the updated algorithm is shown **in the common response**.
>
> ***5. A typo: in line 213, after comma, "we" instead of "We" (wrong capitalized "W")***
>
> Thanks for pointing out this typographical error. We will correct it in the revised version.
>
> ***6. Why there is no SVD+RGB feedback in the ablation study? I think that is important to see how does the coordinates map contribute compared to multi-view RGB images as feedback.***
>
> We apologize for our oversight. We **have included** the qualitative and quantitative results of this setting in **Fig.6 and Tab.1 of our attached PDF**. The experimental results confirm the necessity of our two feedback types, demonstrating that CCM feedback enhances detailed geometry, while RGB feedback contributes to high-quality texture. The combination of  these two feedback types brings greater benefits.

---

> > ### Comment · Reviewer_DdFX · 2024-08-13
> >
> > Thank you for providing a detailed rebuttal and for conducting additional experiments and the new ablation study, which demonstrates how integrating RGB and CCM images from 3D reconstructions into the 2D diffusion process can enhance reconstruction capabilities.
> >
> > While the improvement in reconstruction ability is appreciated, my primary concern regarding the innovative contribution of the proposed approach remains unaddressed. The core concept of utilizing 2D multiview diffusion supplemented with 3D-aware features closely mirrors methodologies already explored in SyncDreamer, where a similar technique of constructing a 3D feature volume and projecting it to each view as an additional 3D-aware condition is applied. Additionally, the 3D reconstruction model employed closely resembles the sparse view reconstructor used in LGM, further diluting the distinctiveness of the proposed method.
> >
> > The idea of combining a 3D reconstruction model to provide 3D-aware features for a 2D multiview generation process is conceptually sound and holds potential. However, for this approach to make a significant impact and to truly advance the field, it requires further development to distinguish it from existing methods and to deepen its methodological insights.
> >
> > In light of these considerations, I have decided to maintain my initial evaluation score. I believe that with significant refinements, particularly in developing unique aspects and insights beyond those provided by SyncDreamer and LGM, the proposed method could potentially offer a noteworthy contribution to the field.

---

> ### Author Response · Authors · 2024-08-13
>
> Thank you for your feedback and for bringing these concerns. Our previous response to your questions may have led to your misunderstanding of the core contribution of our approach. We would like to clarify the distinct contributions and the unique value of our method.
> - **Comparison with SyncDreamer.** In terms of specific practices, our 3D-aware feedback differs from the volume features used in SyncDreamer[1]. While many multi-view generation methods, such as MVDream[2], EpiDiff[3], and SPAD[4], employ "3D-aware" features to enforce multi-view consistency, the 3D information they use does **not derive from explicit and accurate 3D models.** Instead, these methods typically rely on implicit multi-view interactions—like full multi-view self-attention in MVDream, epipolar-constraint attention in EpiDiff and SPAD, or SyncDreamer’s use of latent volumes and projection features. However, these approaches do not accurately reflect the true 3D geometric structure. **Our SC3D framework focuses on explicitly modeling 3D geometric structures with more accurate 3D-aware information derived from the reconstruction module.** Moreover, the explicit geometric structure provided by our SC3D framework allows it to be adaptable and generalizable across various network designs other than the implicit geometric features that should be trained or fine-tuned in each applied scenario.
> - **Discussion about LGM.** The SC3D framework is designed to seamlessly integrate multi-view generation with reconstruction, demonstrating robust adaptability across different models. While the LGM[5] model is employed in our reconstructions, SC3D's inherent versatility allows it to also **incorporate alternative models** such as GRM[6], GS-LRM[7], among others. This flexibility enhances the framework's ability to adapt to a wide range of reconstruction models, distinguishing it from more rigid systems that are tightly coupled to specific models.
> - **Our key contributions.** Our SC3D **framework** integrates multi-view generation with 3D reconstruction, aiming to reduce the "data bias" between these two stages. To address these challenges,  we implement a self-conditioning mechanism and employ a joint training approach. Our multi-view generator in our framework models explicit 3D geometric features by incorporating **accurate 3D-aware** information from the reconstruction model. Our framework is **highly extensible** and can accommodate various multi-view generation networks and reconstruction networks.
>
> We sincerely hope that you will reconsider the contributions and value of our approach. Looking forward to your feedback again.
>
> [1] SyncDreamer: Generating Multiview-consistent Images from a Single-view Image (ICLR 2024)
>
> [2] MVDream: Multi-view Diffusion for 3D Generation (ICLR 2024)
>
> [3] EpiDiff: Enhancing Multi-View Synthesis via Localized Epipolar-Constrained Diffusion (CVPR 2024)
>
> [4] SPAD : Spatially Aware Multiview Diffusers (CVPR 2024)
>
> [5] LGM: Large Multi-View Gaussian Model for High-Resolution 3D Content Creation (ECCV 2024)
>
> [6] GRM: Large Gaussian Reconstruction Model for Efficient 3D Reconstruction and Generation
>
> [7] GS-LRM: Large Reconstruction Model for 3D Gaussian Splatting

---

### Official Review · Reviewer_RUZS · 2024-07-12

**Soundness:** 3
**Presentation:** 3
**Contribution:** 3
**Rating:** 6
**Confidence:** 4

**Summary:**

This paper proposes a method for 3D asset generation conditioned on a single image. The approach follows the recent trend of a two-stage feed-forward model – first generating multi-view images and then using a sparse-view reconstructor to reconstruct the 3D object (specifically, LGM in this paper). This two-stage model has a significant drawback: the inconsistency of the multi-view generation model may result in an imperfect input for the reconstructor, thus causing quality degradation of the final generated 3D assets.

To address this issue, the authors propose adding a 3D-aware feedback mechanism to improve multi-view consistency and enhance the final reconstructed results. Specifically, a self-conditioned mechanism is introduced, where the output of the reconstruction model is fed into the diffusion model. This output is involved in the diffusion process, leading to better 3D consistency.

Overall, the method seems sound to me.

**Strengths:**

(1) The problem definition and the motivation for the project are very clear.

(2) The paper is well-written and easy to understand.

(3) The method seems sound. By adding the rendering results of a reconstructor as input, which present strong multi-view consistency, the diffusion model is also capable of generating multi-view-consistent images.

(4) Table 3 appears reasonable and as expected.

(5) The appendix provides helpful details on training and network architectures, aiding in the reproduction of the results.

**Weaknesses:**

(1) Some related works lack citation and discussion:
(a) In "Dmv3d: Denoising Multi-View Diffusion Using 3D Large Reconstruction Model" [ICLR 2024], the paper uses a similar mechanism (though not entirely the same) by employing a 3D reconstructor as a multi-view image denoiser.
(b) “Carve3D: Improving Multi-View Reconstruction Consistency for Diffusion Models with RL Finetuning” [CVPR 2024] enhances multi-view consistency through RL fine-tuning.

(2) I encourage the authors to provide more visual results to help readers understand and appreciate the diffusion/reconstruction process. For example, could the authors provide some visual results of $\tilde{x}_0$ at different denoising steps?

(3) In the comparisons, although quantitative results are provided, could the authors include some qualitative (visual) comparisons to the baseline methods?

Other minor issues:

(1)	Line 118, The Plucker coordinate should be (d, o x d)

(2)	Line 206, We -> we

(3)	Line 225, meshe -> meshes

**Questions:**

Not applicable

**Limitations:**

(1)	The method is limited to object-level reconstruction with a clean background. Though this is a common limitation in recent related works, I encourage the authors to explore this issue in future work.

(2)	As discussed in the paper, extracting high-quality surface geometry from the Gaussian model remains an open problem. This is an interesting topic for future research.

---

> ### Author Rebuttal · Authors · 2024-08-07
>
> ***1. Some related works lack citation and discussion: DMV3D and Carve3D.***
>
> Thanks for your suggestion, and we will cite DMV3D and Carve3D with more discussions in our revised paper:
> - DMV3D employs a 3D reconstruction model as the 2D multi-view denoiser in a multiview diffusion framework, to achieve generic end-to-end 3D generation. However, it does not leverage the powerful capabilities of pretrained image or video diffusion models, and training from scratch on 3D data limits its generalization. In contrast, our unified framework can leverage the visual prior from large image and video datasets, offering greater potential and capacity to tackle complex 3D modeling challenges.
> - Carve3D employs a RLFT algorithm with a multi-view consistency metric to enhance the consistency of multi-view diffusion models. This metric is computed by comparing the generated multi-view images with those rendered from the reconstructed NeRF. However, Carve3D does not address the challenge of poor reconstruction quality resulting from limited and differently distributed training data for reconstruction model, which is the second argument for "data bias" we put forward in line 32 of the original paper.
>
> ***2. I encourage the authors to provide more visual results to help readers understand and appreciate the diffusion/reconstruction process. For example, could the authors provide some visual results of x~0 at different denoising steps?***
>
> Thanks for the helpful advice. We visualize reconstruction results at different denoising steps in **Fig.5 of the attached PDF**. Results show that floaters and distorted geometries are generated in the early stages due to multi-view inconsistency. The quality of geometry and appearance tends to be higher in denoising process.
>
> ***3. In the comparisons, although quantitative results are provided, could the authors include some qualitative (visual) comparisons to the baseline methods?***
>
> We add more visual comparisons to the baseline method and results in **Fig.3 and Fig.4 of our attached PDF**. We compare our SC3D with SOTA image-to-multiview generation methods and image-to-3D generation methods. The results show that our SC3D generates more consistent and higher-quality multi-view images, and produces 3D assets with superior geometry and textures. We provide our detailed analysis **in the common response**.
>
> ***4. Other minor issues about typos.***
>
> Sorry for our carelessness. We will fix it in the revised version.

---

### Official Review · Reviewer_VBi9 · 2024-07-12

**Soundness:** 3
**Presentation:** 2
**Contribution:** 3
**Rating:** 6
**Confidence:** 3

**Summary:**

The paper observes that the current state-of-the-art image-to-3D generation models consist of two separate parts: generate multi-view images from a single image and run on top the 3D reconstruction. This process has no feedback loop, i.e. the reconstruction does not inform the image generation which in turn leads to a worse quality of reconstruction. They propose a method that builds in a feedback to loop back the feedback of the reconstruction into the diffusion process. They report superior 3D reconstruction quality over the usual two separate step method.

**Strengths:**

1. The overall idea that drives the paper to provide a link between 3D reconstruction and diffusion at training and inference time is very powerful and novel, and has not been explored in existing text-to-3D papers. I think this is a significant asset of the paper in a crowded area.
2. The results presented seem to improve quite a bit over the existing state-of-the-art for the results shown.

**Weaknesses:**

1. The presentation of the paper is not clear.
    - In Fig. 1 the paper is describing an iterative process. Fig. 1 has also no output, but suggests the 3D representation is the output. However, Fig. 2 suggests the (multi-view) images are the final output? Are the two decoders the same? In the paper you are referring to different models G and F. They are not mapped to the figure to get a better picture.
    - Lines 169-172: this is describing the training strategy. That should be moved to the part starting from line 180 where you are actually describing the training strategy.
    - Equation 1: c_skip is not explained
    - The paper has many typos. Especially in part 4, they appear in almost every paragraph.
    - Lines 227-228: This statement seems contradicting: “Directly employing a NeRF-based feed-forward model during the training process significantly reduces training speed due to the computational demands of volumetric rendering.”
    - Replacing the algorithm code with more concise pseudo code may make it much easier for more readers to understand.
2. Comparisons are not very comprehensive
    - None of the methods in Figure 1 are qualitatively compared.
    - In Figure 3, it seems different views are compared in the first and second column
    - Minor: It would be also really helpful to introduce some visual cues into figure 9 to easier grasp the results.
3. Some claims are not justified
    - Is the section on augmenting the diffusion model with camera control in 3.1 a claimed contribution of the paper? The statement that: “This approach allows for more detailed and accurate 3D rendering, as pixel-specific embedding enhances the model’s ability to handle complex variations in depth and perspective across the video frames.” Is not justified at all, as other types of embeddings are not ablated.

**Questions:**

My questions are described in the weaknesses section, but the most important is fleshing out the comparisons to understand what sort of improvement this method makes. The quantitative comparisons are dependent on the generated multi-view images to compare with for PSNR, so the reconstructed results aren’t necessarily good, but rather just match what the model generated. Having qualitative comparisons to existing methods would go a long way in answering my question of whether or not the proposed method is an improvement.

**Limitations:**

Overall the paper does describe some limitations of the method, but it’s not clear if they are relevant. For example, is using the gaussian splatting method really a limitation in this case? I’d be interested to know how long this method takes (is it much slower than LGM baseline), how computationally intensive it is, or how sensitive it is to the initial generation by the multi-view diffusion model.

---

> ### Author Rebuttal · Authors · 2024-08-07
>
> ***1. The presentation of the paper is not clear.***
>
> - *Confusion about Fig.1 and Fig.2 in the original paper.* We slightly adjust these two figures to ensure that both outputs are 3D representations (The updated figure sees **Fig.1 in the attached PDF**.). The two decoders are the same VAE decoder, and the two reconstruction models are the same too. We also include symbols $\mathcal{G}$ and $F$ in the updated figure.
> - *Move the lines 169-172 to where we actually describe the training strategy.* We agree with the suggestion and the mentioned lines are moved to the "Training Strategy" section in the revised version.
> - *Explanation of c_skip in Equ.1.* $c_\text{skip}$ is a parameter that controls how much of the original x_0 is retained. Its form is similar to the skip connection and is defined as $c_\text{skip}(\sigma)=\frac{1}{1+\sigma^2}$. The definition can be found in Appendix A.1 of our original paper.
> - *Typos.* We committ to fixing these errors, and have conducted a thorough proofreading of the manuscript.
> - *Contradictory statement in lines 227-228.* Sorry that our description causes some misunderstanding. The statement aims to show that the low speed of volumetric rendering makes it challenging to replace our current Gaussian Splatting reconstruction model with a NeRF-based reconstruction model. We will make it clearer.
> - *Replacing the algorithm code with more concise pseudo code.* We provide the pseudo code **in the common response**, and will update the manuscript.
>
> ***2. Comparisons are not very comprehensive.***
>
> - *None of the methods in Figure 1 are qualitatively compared.* We include more comprehensive qualitative comparisons **in our attached PDF**. As shown in **Fig.4**, the two-stage baseline methods like LGM and InstantMesh, often generate incorrect or incomplete geometry due to the inconsistency of the generated multi-view images. We also conduct ablation studies on our feedback machinism in **Fig.6**. The reconstructed results show that our SC3D with full 3D-aware feedback achieves superior performance in both geometry and texture. Please check **the common response** for detailed analysis and visualization.
> - *Comparison of different views in Figure 3.* We apologize for the confusion and fix it in **Fig.4 of the attached PDF**.
> - *Add visual cues in Figure 9.* We appreciate the suggestion and will add visual cues as we did in our attached PDF.
>
> ***3. Some claims are not justified.***
>
> - *Is the section on augmenting the diffusion model with camera control in 3.1 a claimed contribution of the paper?* The usage of Plücker embedding is not one of our paper’s contributions. It is a common positional encoding utilized in several recent works [1~4]. Our approach simply adopts this design.
>
> [1] DMV3D: Denoising Multi-View Diffusion Using 3D Large Reconstruction Model
>
> [2] SPAD: Spatially Aware Multi-View Diffusers
>
> [3] EpiDiff: Enhancing Multi-View Synthesis via Localized Epipolar-Constrained Diffusion
>
> [4] Free3D: Consistent Novel View Synthesis without 3D Representation
>
> ***4. My questions are described in the weaknesses section, but the most important is fleshing out the comparisons to understand what sort of improvement this method makes. Having qualitative comparisons to existing methods would go a long way in answering my question.***
>
> We've provided our detailed responses in the above reply and the common response. In summary, we conduct detailed visual comparisons in **Fig.3, Fig.4 and Fig.6 of our attached PDF**. Qualitative comparisons for image-to-multiview (Fig.3) and image-to-3D (Fig.4) generation show that our SC3D generates more consistent and higher-quality 3D assets than other methods. The ablation study on feedback (Fig.6) shows the importance of our feedback loop in improving texture quality and geometric details. We provide the detailed analysis in **the common response**.
>
> ***5. Overall the paper does describe some limitations of the method, but it’s not clear if they are relevant. For example, is using the gaussian splatting method really a limitation in this case? I’d be interested to know how long this method takes (is it much slower than LGM baseline), how computationally intensive it is, or how sensitive it is to the initial generation by the multi-view diffusion model.***
>
> - *Is using the GS method a limitation?* We consider that 3D mesh is more commonly used in downstream applications, and there are still challenges in converting Gaussian Splatting to high-quality meshes.
>
> **Table B**
> |Model|Inference time|
> |-|-|
> |LGM baseline (ImageDream + LGM)|1.225s|
> |SV3D + LGM|24.18s|
> |SC3D (SVD + LGM + Feedback) |25.19s|
>
> - *Computational cost.* We acknowledge that our SC3D has limitations in computation cost, and will include it in the revised version. In **Tab.B**, we compare the inference time of baseline methods under the same setting. The LGM baseline employs ImageDream[5] to generate 4 views of 256x256 resolution, and reconstruct them into 3DGS. Our SC3D approach uses SVD to generate 8 views of 512x512 resolution. **For a fair comparison**, we report the inference time of "SV3D + LGM", where SV3D[6] is a multi-view generator fine-tuned from SVD. Compared to "SV3D + LGM", our additional overhead mainly arises from the feedback mechanism at each step, involving VAE decoding, 3D reconstruction and rendering, and conditioning injection. As shown in **Tab.B**, the inference time remains within acceptable bounds.
> - *Sensitivity to initial multi-view generation.* SC3D is not sensitive to the initial multiviews. Our reconstruction model may produce poor results in the early steps due to the inconsistency of the initial multi-view images. But the quality of the reconstructed results rapidly increases in the denoising process (see **Fig.5 in the attached PDF**).
>
> [5] ImageDream: Image-Prompt Multi-view Diffusion for 3D Generation
>
> [6] SV3D: Novel Multi-view Synthesis and 3D Generation from a Single Image using Latent Video Diffusion

---

> > ### Comment · Reviewer_VBi9 · 2024-08-13
> >
> > Thank you for your detailed rebuttal. The additional comparisons and ablations are very insightful, and have increased my perception of the paper. Thus, I am increasing my score. However, the paper will require a significant amount of improvement in the presentation, which makes me less certain that it is ready for publication now.

---

> > > ### Author Response · Authors · 2024-08-14
> > >
> > > We appreciate your feedback and recognition of the improvements in our revised submission. We've revised the figures for clarity, improved textual organization and clarity, provided additional technical explanations, and thoroughly proofread the manuscript to enhance overall quality. Thank you for your guidance in improving our submission!

---

### Official Review · Reviewer_KsJi · 2024-07-13

**Soundness:** 4
**Presentation:** 3
**Contribution:** 3
**Rating:** 5
**Confidence:** 5

**Summary:**

This paper proposes SC3D for the single-image-to-3D generation, which integrates the diffusion-based multi-view generation and Gaussians-based 3D reconstruction through a self-conditioning mechanism. Specifically, during each denoising step, SC3D injects the rendered image and geometric map from the reconstruction model into the denoising process to enhance the multi-view consistency of the multi-view generated images. Experiments on GSO dataset demonstrates its superiority over existing methods mentioned in this paper.

**Strengths:**

1. SC3D integrates multi-view image generation and 3D reconstruction into a single framework, ensuring similar data distribution between the two modules and thereby improving reconstruction quality during the reference process.

2. SC3D proposes a self-conditiond 3D-award feedback mechanism to bridge the multi-view image generation and 3D reconstruction, in which the rendered images and geometric map are injected in the multi-view generation network. Such design makes sense and could improve the consistency of the generated results from the multi-view generation network.

**Weaknesses:**

1. Lack detailed visual comparisons with baseline methods. The authors only compare SC3D with LGM but do not show results generated from other baselines, making the visual comparison results less convincing.

2. The paper suffers from poor organization. For example, Figure 4 and Figure 5 are not referenced anywhere in the text. The purpose of Figure 6 is confusing, as its caption suggests it shows results from another work, and it is difficult to discern differences among the three rows. Additionally, the paper's typesetting is of poor quality. There are many blank spaces in the text.

**Questions:**

1. Will jointly training the multi-view generation network and 3D reconstruction network make the training unstable? Will the jointly training mechanism increase the training time? More details about potential disadvantages of jointly training should be discussed in the paper.

2. As mentioned in Line 211-218, the setting of two ablated experiments seems same, what is the difference between them?

**Limitations:**

Please refer the weaknesses and questions above.

---

> ### Author Rebuttal · Authors · 2024-08-07
>
> ***1. Lack detailed visual comparisons with baseline methods.***
>
> We conduct comparisons with more baseline methods, and show the visualization results in **Fig.3 and Fig.4 of the attached PDF**. We compare SC3D with the SOTA image-to-multiview and image-to-3D generation methods.
> - *Image-to-multiview generation.* We compare SC3D with SyncDreamer[1], SV3D[2] and VideoMV[3], and present the qualitative results in **Fig.3 of our attached PDF**. SyncDreamer and SV3D fine-tune image or video diffusion models on 3D datasets but do not use explicit 3D information, often resulting in blurry textures or inconsistent details. VideoMV aggregates rendered views from reconstructed 3D models at the inference stage, but it fails to take into account the "data bias" between these two stages. Although VideoMV improves the multi-view consistency, it introduces biased information from reconstructed 3D models, leading to results that are unaligned with the input image. Our SC3D framework involves the joint training of the two stages and uses geometry and appearance feedback for multi-view generation, enhancing the consistency and quality of the generated multi-view images.
> - *Image-to-3D generation.* We compare SC3D with TripoSR[4], VideoMV[3], LGM[5] and InstantMesh[6], with visualization results in **Fig. 4 of the attached PDF**. TripoSR reconstructs 3D model from a single image without using generative models, resulting in low-quality geometry/appearance and limited generalizablity. VideoMV reconstructs 3DGS from its generated multi-view images. Due to its biased multiview generation (as indicated in Fig.3 in the attached PDF), it may generate inconsistent texture against the input image and somewhat distorted geometry. Moreover, the two-stage baseline methods like LGM and InstantMesh (i.e., an off-the-shelf image-to-multiview generation method plus LGM or InstantMesh to fulfil the image-to-3D generation) produce incomplete or inconsistent geometry due to the gap between the two stages. Our SC3D bridges the multiview generation and 3D reconstruction where the benefits of each module can be transferred to the other one, therefore generating high-quality 3D assets.
>
> [1] SyncDreamer: Generating Multiview-consistent Images from a Single-view Image (ICLR 2024)
>
> [2] SV3D: Novel Multi-view Synthesis and 3D Generation from a Single Image using Latent Video Diffusion (Arxiv 2403.12008)
>
> [3] VideoMV: Consistent Multi-View Generation Based on Large Video Generative Model (ECCV 2024)
>
> [4] TripoSR: Fast 3D Object Reconstruction from a Single Image (Arxiv 2403.02151)
>
> [5] LGM: Large Multi-View Gaussian Model for High-Resolution 3D Content Creation (ECCV 2024)
>
> [6] InstantMesh: Efficient 3D Mesh Generation from a Single Image with Sparse-view Large Reconstruction Models (Arxiv 2404.07191)
>
> ***2. The paper suffers from poor organization.***
>
> We acknowledge the issues with our paper's organization and typesetting, and will seriously polish the manuscript.
> - *Figures 4 and 5 are currently not referenced in the text.* In the revised version, we will ensure that these figures, which illustrate the comparison with baseline methods and out-of-distribution (OOD) testing results, are properly integrated into Section 4 to strengthen our analysis and conclusions.
> - *The purpose of Figure 6 is confusing, as its caption suggests it shows results from another work.* We included Figure 6 in the original version to show that VideoMV's fusion of reconstruction results at the inference stage leads to a deviation in appearance from the input. To avoid any confusion, we will remove Figure 6 and provide a clearer and more comprehensive comparison like Fig.3 and Fig.4 in the attached PDF.
> - *Typesetting and Formatting.* We will carefully re-evaluate the typesetting to eliminate blank spaces and enhance the overall layout.
>
> ***3. Will jointly training the multi-view generation network and 3D reconstruction network make the training unstable? Will the jointly training mechanism increase the training time? More details about potential disadvantages of jointly training should be discussed in the paper.***
>
> - *Training stability.* Our joint training method is stable. This benefits from the following two aspects. (We will include additional training details in the revised version.)
>   1. The pretrained video diffusion model effectively utilizes its powerful visual generation capabilities to produce high-quality initial images.
>   2. We initialize the output layers of the condition encoders to zero, ensuring that even suboptimal initial reconstruction results do not adversely affect the network significantly.
> - *Training time.* Jointly training the multi-view generation network and the 3D reconstruction network increases training time and requires more GPU memory, as it involves training two models simultaneously with the feedback mechanism. We've measured the time required for 1,000 training steps on a single A100 GPU with the setting listed in Appendix B of our paper, as detailed in **Tab.A**. We will further discuss the computation requirements in Section 4.3 (Limitations) of the revised paper. It is worth noting that SC3D has minimal impact on inference speed, as the 3D feedback mechanism incurs only a slight overhead.
>
> **Table A**
> | Setting | Training Time |
> |-|-|
> | train multi-view diffusion model only (SVD) | 15 min |
> | train reconstruction model only (LGM) | 10 min |
> | train SC3D (SVD + LGM + Feedback) | 36 min |
>
> ***4. As mentioned in Line 211-218, the setting of two ablated experiments seems the same, what is the difference between them?***
>
> Sorry for the confusion of the notations in the table. Actually, in the "Variant" column, "SVD" means the metrics for multi-view generation results, while "GS" shows the metrics for the reconstructed 3D results. In the revised version, we will merge Tab.2 and Tab.3 from the original paper to provide a clearer comparison. The updated table refers to **Tab.1 in our attached PDF**.

---

> > ### Comment · Reviewer_KsJi · 2024-08-13
> > **Official Comment by Reviewer KsJi**
> >
> > Thanks for providing the detailed rebuttal and additional experiments. My concerns are addresses. I will change my rating to Borderline accept.

---

### Author Rebuttal · Authors · 2024-08-07

We thank all reviewers for the constructive comments and for recognizing the novelty and effectiveness of our self-conditioned image-to-3D generation method with 3D-aware feedback. We also extend our gratitude to the reviewers for identifying shortcomings in our paper's presentation and organization. We will seriously make revisions to enhance its readability and presentation. Here, we provide responses to common questions raised by reviewers.
- **Generalizability.** As shown in **Fig.2 of our attached PDF**, SC3D has strong generalizability, generating high-quality 3D assets from out-of-distribution images, including real world images.
- **Visual comparison with baseline methods.** Our SC3D can generate multi-view images and 3D models that are consistent with each other. To further assess its effectiveness, we compare SC3D with the SOTA image-to-multiview and image-to-3D generation methods.
  - **Image-to-multiview generation.** We compare SC3D with SyncDreamer[1], SV3D[2] and VideoMV[3], as shown in **Fig.3** of the attached PDF. SyncDreamer and SV3D fine-tune image or video diffusion models on 3D datasets but lack explicit 3D information, often resulting in blurry textures or inconsistent details. VideoMV aggregates rendered views from reconstructed 3D models at the inference stage but fails to take into account the "data bias" between two stages. Although VideoMV improves the multi-view consistency, it introduces biased information from the reconstruction stage, leading to results that are unaligned with the input image. Our SC3D uses joint training of the two stages and uses geometry and appearance feedback for multi-view generation, generating consistent and high-quality multi-view images.
  - **Image-to-3D generation.** We compare SC3D with TripoSR[4], VideoMV[3], LGM[5] and InstantMesh[6], as visualized in **Fig.4** of the attached PDF. TripoSR struggles with high-quality geometry and appearance due to lacking large pre-trained generative models. VideoMV reconstructs 3DGS from its generated multi-view images, but its inherent biases in multiview generation can lead to misaligned textures and distorted geometries. Two-stage methods such as LGM and InstantMesh(comprising an off-the-shelf image-to-multiview generation method followed by reconstruction models for the image-to-3D generation process), often yield incomplete geometry due to the disparity between multiview generation and 3D reconstruction. In contrast, our SC3D framework integrates multiview generation and 3D reconstruction, enhancing each module's strengths to produce high-quality 3D assets.
- **Performance analysis.** To effectively analyze our method's effectiveness, we visualize the generation process and conduct comprehensive ablation experiments.
  - **Denoising process.**  **Fig.5** of the attached PDF shows the reconstruction results at various denoising steps, demonstrating our self-conditioning denoising process. The geometry and texture of objects progressively improve through iterative refinement.
  - **Ablation study.** We've reorganized the ablation study and the presentation (see qualitative results in **Fig.6** and quantitative results in **Tab.1** of the attached PDF). Visualization results show that the baseline without feedback generates low-quality and inconsistent results. Using only coordinates map feedback results in blurry textures, while only RGB feedback leads to poor geometric details. Combining both significantly enhances geometry and texture quality. We've alse added an ablation experiment using only RGB and merged the original tables into **Tab.1** in the attached PDF to for better comparison. Quantitative results show that feedback using both RGB and coordinates map achieves superior outcomes. Furthermore, our framework reduces the performance gap between the generated multi-view images and 3D representation, enhancing overall performance.
- **Presentation and organization.** We take the issues with our paper's organization and typesetting seriously and are committed to improving the manuscript to enhance its readability and presentation.
- **More concise pseudo code.** Our revised pseudo code is shown as Algorithm 1 and Algorithm 2.
```
Algorithm1 Train_loss
Input: x, cond_image, cameras, timestep
Output: loss
Description: Returns the loss on a training example x. Details about EDM are omitted here.

Begin
    noise <- Sample from Normal Distribution
    noisy_x <- Add_Noise(x, noise, timestep)
    pred_x <- F(noisy_x, cond_image, timestep, cameras)
    pred_i <- VAE_Decoder(pred_x)
    self_cond <- G(pred_i, cameras, timestep)

    if Random_Uniform(0, 1) > 0.5 then
        pred_x <- F(noisy_x, cond_image, timestep, cameras, self_cond)

    loss_mv <- MSE_Loss(pred_x, x)
    loss_recon <- MSE_Loss(self_cond, x) + LPIPS_Loss(self_cond, x)
    loss <- loss_mv + loss_recon

    Return loss
End
```
```
Algorithm2 Inference
Input: cond_image, cameras, timesteps
Output: images, 3d_model
Description: Generate multi-view images and 3D model from a condition image.

Begin
    self_cond <- None
    x_t <- Sample from Normal Distribution
    for each timestep in timesteps do
        pred_x <- F(x_t, cond_image, timestep, cameras, self_cond)
        pred_i <- VAE_Decoder(pred_x)
        self_cond <- G(pred_i, cameras, timestep)
    End For

    Return pred_i, self_cond
End
```

[1] SyncDreamer: Generating Multiview-consistent Images from a Single-view Image (ICLR 2024)

[2] SV3D: Novel Multi-view Synthesis and 3D Generation from a Single Image using Latent Video Diffusion (Arxiv 2403.12008)

[3] VideoMV: Consistent Multi-View Generation Based on Large Video Generative Model (ECCV 2024)

[4] TripoSR: Fast 3D Object Reconstruction from a Single Image (Arxiv 2403.02151)

[5] LGM: Large Multi-View Gaussian Model for High-Resolution 3D Content Creation (ECCV 2024)

[6] InstantMesh: Efficient 3D Mesh Generation from a Single Image with Sparse-view Large Reconstruction Models (Arxiv 2404.07191)

---

### Decision · Program_Chairs · 2024-09-25

**Decision:**

Reject

**Comment:**

The paper received four reviews from expert reviewers. The rebuttal and discussion have cleared up many concerns about structure, content, clarity, comparisons, and visual results. This has led to a slight increase in the overall rating of the paper.
However, the promised changes needed to bring the paper to the level of NeurIPS go beyond what can be accepted without another review. The AC find the additional results, clarifications, and comparisons essential, as well as the promised improvements in writing, structure, typos, and other fixes and encourages the authors to resubmit a revised version to the next suitable venue.